# Acute Lymphoblastic Leukemia Immunotherapy Treatment: Now, Next, and Beyond

**DOI:** 10.3390/cancers15133346

**Published:** 2023-06-26

**Authors:** Anna Aureli, Beatrice Marziani, Adriano Venditti, Tommaso Sconocchia, Giuseppe Sconocchia

**Affiliations:** 1CNR Institute of Translational Pharmacology, Via Carducci 32, 67100 L’Aquila, Italy; 2Emergency Medicine Department, Sant’Anna University Hospital, Via A. Moro, 8, Cona, 44124 Ferrara, Italy; mrzbrc@unife.it; 3Department of Biomedicine and Prevention, The University of Rome “Tor Vergata”, 00133 Rome, Italy; adriano.venditti@uniroma2.it; 4Division of Hematology, Department of Internal Medicine, Medical University of Graz, 8036 Graz, Austria; tommaso.sconocchia@medunigraz.at

**Keywords:** ALL, immunotherapy, antibody–drug conjugate, CAR-based therapies, targeted therapies

## Abstract

**Simple Summary:**

The recent emergence of targeted therapies, including antibody–drug conjugates, bispecific antibodies, and CD19 chimeric antigen receptor (CAR) T cell therapy, revolutionized B-lineage acute lymphoblastic leukemia (B-ALL) management, allowing certain optimism, at least for adult patients with Ph+ B-ALL, on gradually replacing chemotherapy and hematopoietic stem cell transplantation in the first remission. However, to date there are still too few patients that benefit from these new therapies. Therefore, future research directions aim to improve the life expectancy of every patient and especially of those with ALL resistant to available therapeutic strategies. This review provides an overview of new treatment paradigms being used in the relapsed/refractory setting as well as current trials through which these new therapies might be introduced to the frontline setting.

**Abstract:**

Acute lymphoblastic leukemia (ALL) is a blood cancer that primarily affects children but also adults. It is due to the malignant proliferation of lymphoid precursor cells that invade the bone marrow and can spread to extramedullary sites. ALL is divided into B cell (85%) and T cell lineages (10 to 15%); rare cases are associated with the natural killer (NK) cell lineage (<1%). To date, the survival rate in children with ALL is excellent while in adults continues to be poor. Despite the therapeutic progress, there are subsets of patients that still have high relapse rates after chemotherapy or hematopoietic stem cell transplantation (HSCT) and an unsatisfactory cure rate. Hence, the identification of more effective and safer therapy choices represents a primary issue. In this review, we will discuss novel therapeutic options including bispecific antibodies, antibody–drug conjugates, chimeric antigen receptor (CAR)-based therapies, and other promising treatments for both pediatric and adult patients.

## 1. Introduction

ALL is a hematologic malignancy characterized by the uncontrolled proliferation of early lymphoid precursors that infiltrate bone marrow [1,2,3].

The central nervous system (CNS) and testes are the most common sites of precursors’ extra-medullary spread [4], although theoretically, any organ or tissue could be infiltrated. The involvement of skin, kidneys, and ovaries has also been extensively described [5,6].

ALL is divided into tumors of B-lineage, T-lineage, and uncommon variants of NK cell lineage which are morphologically indistinguishable. According to the 2022 revisions to the World Health Organization (WHO) and International Consensus Classification (ICC), the classification of major subtypes of ALL includes four distinct entities: B-ALL/LBL not otherwise specified (NOS), B-ALL/LBL with recurrent genetic abnormalities, T-ALL/LBL, and NK-ALL/LBL [7,8], as shown in Table 1.

Few environmental and/or genetic factors have been associated with an increased risk of ALL. Among these, ionizing radiation, pesticide exposure, childhood infections [9,10,11], and genetic conditions such as Down syndrome or ataxia telangiectasia are included [12,13,14,15].

Its incidence varies among people of different ages, sex, and race [16,17,18]. The age-specific incidence curve for ALL has a bimodal distribution with peak incidences in children aged between 1 and 4, and adults aged 55 or above [19]. Males develop it more than females with a ratio of 1.2:1 [20].

Globally, the estimated annual incidence of ALL is 1 to 5 cases/100,000 population, and more than two-thirds of cases of ALL are of the B-cell phenotype [17,21,22,23]. Italy, the USA, Switzerland, and Costa Rica are the countries with the highest ALL incidence [17]. In the USA, about 6660 new cases and 1560 deaths (including both children and adults) were estimated in 2022 [20].

The outcome is more disappointing in adults (5-year overall survival (OS) < 45%) than in children (5-year survival rate of over 90%) [24,25,26], and this is related to multiple factors such as a higher incidence of poor prognostic markers, a lower incidence of favorable subtypes, and traditional chemotherapy regimens [27].

Although there has been a substantial improvement in OS over time, there is still a gap in the availability of leukemia treatments between countries. This is partly due to the different socioeconomic status; low-income countries are less likely to use available treatments and this may contribute to poor survival [28].

Therefore, there is a joint effort to find more accessible solutions and to develop promising therapeutic strategies aiming to maintain remission, improve survival, and control the toxicities associated with chemotherapy regimens.

## 2. Different Biological Characteristics in Pediatric and Adults ALL Patients

Childhood and adult ALL are biologically distinct and diverge in their molecular landscape but also their cellular origin [1]. Even if the exact causes of ALL are not yet understood, it has been demonstrated that in children, it is the result of a multistep process associated with the acquisition of genetic alterations in lymphoid progenitors during inutero development [29,30]. Chromosome aneuploidy, structural alterations, rearrangements, copy number variations (CNVs), and sequence mutations all contribute to leukemogenesis [31].

Disease cytogenetic abnormalities have a different prognostic impact between age categories. Usually, adult patients have a higher white blood cell count, an increased frequency of T-lineage ALL, and a decreased incidence of hyperdiploidy than children [32,33]. Also demonstrated was an increase in the presence of unfavorable genetic anomalies with increasing age (incidence up to 53% over 55 years), such as the Philadelphia chromosome [34]. In contrast, genetic alterations, such as hyperdiploid karyotype, frequently seen in pediatric ALL patients, are related to a favorable outcome [35].

The gap in outcome between children and adults is due to the differences in disease biology and treatment tolerance and also to the intensified chemotherapy regimens used in children that permit improved response rates and prolonged survival [36]. Fortunately, the management of ALL in adult patients has significantly improved thanks to the administration of pediatric-inspired regimens or even unmodified pediatric protocols (adults up to 60 years old), so chemotherapy intensity has increased [37].

## 3. Evolution of ALL Treatment Applications

Treatment for ALL is divided into four different phases: remission induction, consolidation, intensification, and long-term maintenance. CNS prophylaxis is given at the proper intervals during the treatment. Allogeneic HCT is optional after consolidation.

Standard frontline chemotherapy is used for induction therapy, while targeted drug therapy, alone or combined with chemotherapy, is employed for all phases.

The achievement of the current treatment modalities is the result of changes that happened in a temporal space that started in the 1970s when the older strategies were applied [38].At that time, cranial radiotherapy (CRT) to prevent CNS relapse was used for all patients [39,40]. However, intensive and prolonged therapy for ALL was considered responsible for detrimental effects on intellectual and learning abilities [41,42,43]. As a result, some years later, CRT intensity has been reduced and intrathecal therapy and high doses of systemic chemotherapy substituted the previous method [44,45,46,47,48].

Then, conventional chemotherapy was optimized, raising the chance of cure in the highest-risk patients while minimizing long-term adverse events in those with the lowest risk [49,50,51,52].

However, in childhood ALL, CNS-directed prophylaxis remains an obliged choice. Indeed, the high possibility of infiltration of the CNS, by massive numbers of leukemic cells, puts patients at a higher chance of CNS relapse leading to severe morbidity and mortality [53,54].

Until now, little information is known about where leukemia cells reside in the CNS and about their interactions with cellular components of the CNS microenvironment, which could induce their quiescence and survival. Certain studies suggested that some B-Cell Precursor (BCP)-ALL cells would be able to survive in particular CNS niches for a very long time as extramedullary minimal residue disease and could be responsible for CNS relapse [55,56,57].

Fernandez-Sevilla et al. proposed that the choroid plexus (CP), secretory tissue responsible for producing cerebrospinal fluid, constitutes a sanctuary for pediatric BCP-ALL cells. Inside it, interactions between BCP-ALL cells and microenvironment cell components promote their survival and chemoresistance [58].

Allo-HCT used as a consolidation therapy contributes to the considerable improvement in the prognosis of patients with ALL, but not without complications(complexity and graft vs. host disease (GVHD)). Access to allo-HCT is usually reserved for patients with high-risk characteristics or relapsing disease [59,60,61,62]. Until recently, for those patients as well as for older adults, the treatment options were extremely limited.

## 4. Immunotherapy for ALL

Finding the best treatment for ALL is an ongoing challenge leading to the continuous development of new therapeutic approaches. Among these, immunotherapy stands out, exploiting the patient’s immune system to target cancer cells, improving survival, and reducing the toxicity of chemotherapy. Major immunotherapies include the use of bispecific antibodies, CART or CARNK cells, and antibody–drug conjugates, which are showing important results primarily in the treatment of B-ALL. CART or CARNK cells and antibody therapy also hold promise for the treatment of T-ALL.

### 4.1. Bispecific Antibodies (BsAbs)

BsAbs are antibodies engineered to contain two different fragment-binding antigens (Fabs) regions that allow for the concurrent targeting of two antigens. One of the main mechanisms of action of BsAbs is to recruit and activate effector cells (i.e., T cells) against target cells (i.e., tumor cells). These antibodies, unlike conventional ones, induce enhanced T-cell activation. BsAb-activated T cells exert their cytotoxic action on tumor cells by producing proteins such as perforin and granzymes. Perforin is responsible for pore formation in cell membranes, facilitating the entry of granzymes and thus enabling their delivery into the cytosol to initiate apoptosis [63,64,65].

Blinatumomab is a bispecific T-cell-engaging (BiTE) antibody. BiTE antibodies lack the Fc fragment and are composed of the fusion of two different single-chain variable fragments (scFvs); one scFv binds the CD3 expressed on effector T cells and the other binds a tumor-associated antigen. Blinatumomab, by simultaneously binding CD19 on B-ALL cells and CD3 on T cells, can mediate a direct cross-link between T cells and tumor cells [66,67], resulting in targeted and highly effective tumor cell killing (Figure 1) [68].

It is the first FDA-approved BiTE antibody to treat minimal residual disease (MRD)-positive BCP-ALL, as well as relapsed or refractory (R/R) ALL, both in adult [69,70] and pediatric patients [71].To date, blinatumomab (BLINCYTO^@^)is indicated to treat B-cell precursors in ALL patients. The drug is given under cycles of continuous intravenous infusion at a concentration that depends on the patient’s weight (15 μg/m^2^/day if <45 kg; 28 μg/day if >45 kg). Both relapsed/refractory and MRD-positive patients are eligible for the treatment. The former may receive two induction cycles while the latter are treated with a single induction cycle. The induction cycle(s) is (are) followed by consolidation cycles, up to three cycles both for R/R and for MRD+ patients. Each cycle lasts 28 days, followed by a 14-day break [72]. According to the European Medicines Agency(EMA), the use of blinatumomab in the pediatric population is limited to children aged one or older with Philadelphia chromosome-negative CD19-positive B-cell precursor ALL (CD19+ R/R Ph-negative BCP-ALL), which is refractory or in relapse after receiving at least two prior therapies or in relapse after receiving prior HSCT, or in high-risk first relapse as part of consolidation therapy [73]. Furthermore, as demonstrated in infants with newly diagnosed KMT2A-rearranged ALL, the combination of blinatumomab plus chemotherapy had reliable safety and effectiveness [74]. Cytokine release syndrome (CRS) and neurotoxicity are reported as the main adverse events, most of which can be rapidly resolved. B-ALL patients with R/R or a delayed reduction in MRD after HCT have a severe prognosis. Thus, additional therapeutic strategies need to be identified. The successful implementation of this type of immunotherapy, alongside chemotherapy, may also consider blinatumomab as a useful therapeutic strategy in R/R B-ALL following HCT or as maintenance therapy. Hence, investigators are carrying out clinical studies aimed at understanding whether blinatumomab-based immunotherapy following HCT is feasible, safe, and potentially therapeutic. In 2019, Stain et al. tested blinatumomab in an open-label, single-arm, phase 2 study in adults with Ph-B cell precursors ALL. They observed that the efficacy of blinatumomab in patients with previous allo-HCT resulted in 43% of CR within two treatment cycles. Furthermore, they found CRS, neurotoxicity, and GVHD in 3%, 16%, and 11% of patients, respectively. The conclusion was that blinatumomab is an effective salvage therapy [75]. Subsequently, Gaballa et al. fulfilled a single-center phase 2 clinical trial where four cycles of blinatumomab were given, every 3 months, within the first year following HCT. Grade 4 neutropenia was observed in 19% of patients, while rates of CRS and neurotoxicity were minimal, being 5% (grade 1) and 5% (grade 2), respectively. GVHD grade 2 to 4 was also observed. There were no significant differences in terms of blinatumomab efficiency when the results obtained in the blinatumomab cohort of 21 patients were compared with those of a control group of 36 patients. However, additional studies have identified a subgroup of patients who would benefit from blinatumomab. These patients had the characteristic of producing higher levels of CD8+ effector memory T cells than non-responder controls [76]. In an anticipation phase 1–2, multicenter, non-blinded, non-controlled study, Sakaguchi et al. asked whether maintenance blinatumomab-based immunotherapy, given following allogeneic HCT, is safe and efficient for R/R CD19 + B-LL. Patients who accomplished engraftment and were in CR at 30 days but less than 100 days following HCT were treated with two courses of blinatumomab. The study is ongoing, and no information about complications beyond 1 year is available [77].

Among the three studies cited above, it seems that the administration of blinatumomab following allo-HCT may be feasible. It is still unclear whether it is an effective treatment for B-ALL. Therefore, further studies are required to answer this question. There is no doubt that T cell reconstitution after allo-HCT is very slow and usually follows the reconstitution of NK cells. Thus, close monitoring of T-cell reconstitution in terms of the type and quality of CD8+ T cells is critical for efficient cytotoxic activity [76]. On the other hand, it is known that in some cases, efficient T-cell reconstitution could take about 1 year. Therefore, in case of a delayed reconstitution of efficient CD8+ T cells, it can be compensated by associating blinatumomab with a donor lymphocyte infusion (DLI). However, a negative consequence of this combination could be an increase in toxicity in terms of GVHD, CRS, and neurotoxicity.

#### 4.1.1. Blinatumomab in Adult Patients

The first single-arm study on the use of blinatumomab involved 189 adults with R/R Ph-negative B-ALL. Complete remission/complete remission with partial hematologic recovery (CR/CRh) was achieved in 43% (81/189) of patients within two cycles of blinatumomab with a median OS of 6.1 months [70]. The positive data produced accelerated FDA approval for the use of blinatumomab in adults with R/R Ph-negative B-ALL at the end of 2014. The greater efficacy of blinatumomab compared to conventional chemotherapy was then confirmed in the TOWER study. In this randomized, open-label, multicenter phase 3 trial, 271 out of 495 patients with R/R B-ALL received blinatumomab and 134 patients received standard chemotherapy (2:1 ratio). Patients who received blinatumomab had a better CR rate (34% vs. 16%, *p* < 0.001), greater MRD negativity (76% vs. 48%), and a longer median OS (7.7 vs. 4.0 months; *p* = 0.001) than those treated with chemotherapy.

The efficacy of blinatumomab was also tested in the setting of MRD [78]. In the BLAST study, it was used as monotherapy in adults with MRD-positive B-ALL. For this study, 116 patients were enrolled, and of the 113 evaluable patients, 88 (78%) had a complete MRD response after the first treatment cycle. The median OS was 36.5 months and the relapse-free survival (RFS) was 54% after 18 months of follow-up [72].

Blinatumomab has also been evaluated in patients with Ph-positive B-ALL previously treated with tyrosine kinase inhibitor (TKI)-based therapy (ALCANTARA study). Sixteen patients out of forty-five enrolled achieved a CR/CRh rate of 36% with 88% complete MRD [79].

Furthermore, the Foà et al. study assessed a chemotherapy-free induction and consolidation first-line treatment with dasatinib and blinatumomab for Ph+ ALL in adults. More specifically, the 63 adult patients enrolled in the study underwent treatment with dasatinib plus glucocorticoids, followed by two cycles of blinatumomab [80,81]. In total, 98% of the patients achieved CR and 60% had a molecular response (MR) that further increased up to 81% after the fourth cycle of blinatumomab. At a median follow-up of 18 months, the OS was 95% and RFS was 88%. Finally, 24 patients received HSCT and the transplantation-related mortality was considerably lower (4%) than that of other previous studies. 

#### 4.1.2. Blinatumomab in Pediatric Patients

The blinatumomab investigation in the pediatric population has been encouraging even from the first small case series performed in patients with relapsed ALL after HSCT [82,83]. Since then, worldwide research on the pediatric use of blinatumomab exponentially increased(Table 2) [82,83,84,85,86,87,88,89,90,91,92,93,94,95,96,97,98,99,100,101,102,103,104,105,106,107,108].

Two randomized controlled trials recently published their results. Data from the first trial (NCT02101853) are reported by Brown and colleagues. They described the experience of the Children’s Oncology Group, which conducted a randomized phase 3 clinical trial in the USA, Australia, and New Zealand [104]. Two-hundred and eight children, adolescents, and young adults (aged 1 to 30 years) with B-ALL first relapse were eligible. They received a 4-week reinduction chemotherapy course followed by blinatumomab as post-reinduction consolidation or conventional chemotherapy; HSCT followed both treatments. Improved survival has been detected by substituting intensive chemotherapy with blinatumomab in consolidation therapy. The blinatumomab group showed improved DFS (54.4% vs. 39%), OS (71.3% vs. 58.4%), and MRD clearance (75% vs. 32%) with lower toxicities [104].

Furthermore, results of the trial NCT02393859 were published by Locatelli and colleagues, who showed data obtained using blinatumomab as consolidation therapy instead of chemotherapy before allogeneic HSCT for patients (aged >28 days up to 18 years) with high-risk first-relapse B-ALL. A total of 108 patients were included in this study. Enrollment was terminated early because the prespecified criterion to declare benefit in favor of blinatumomab was met. After a median follow-up time of 22.4 months, the adverse events in the blinatumomab arm were significantly fewer than in the consolidation chemotherapy group (31.5% and 57.4%). Death occurred in 8 blinatumomab-treated patients and 16 chemotherapy-treated ones (14.8% vs. 29.6%). The OS hazard ratio from a stratified Cox proportional hazard model was 0.43 (95% CI, 0.18–1.01). CR was seen in 44 of 49 patients (90%) who were treated with blinatumomab and in 26 of 48 patients (54%) treated with chemotherapy. Treatment with one cycle of blinatumomab before allogeneic HSCT resulted in an improved event-free survival at a median of 22.4 months of follow-up over intensive multidrug chemotherapy [105].

Moreover, in January 2022, Locatelli and colleagues reported the final analysis data of an open-label, single-arm, extended-access international study (RIALTO) conducted across 16 specialized hospitals in Europe and the USA and confirmed the results published in 2020 on the safety and efficacy of blinatumomab. Among 110 patients with CD19-positive R/R BCP ALL (aged >28 days up to 18 years) enrolled in the study, 69 achieved CR within the first two treatment cycles; most of them (73.5%) received an allogeneic HSCT and had better OS compared with those who did not (1-year OS probability: 87% vs. 29%). The authors also referred to a very low incidence of severe CRS (grades 3–4) and grade 3 neurotoxicity arising with blinatumomab treatment. Only two patients developed severe grade ≥3 treatment-related CRS and only six experienced severe grade ≥3 neurologic toxicity. These events resolved quickly (CRS median resolution time, 6.5 days; neurologic toxicity median resolution time, 2.0 days). No adverse events with fatal outcomes were associated with blinatumomab treatment [106].

Recent retrospective studies have also provided evidence supporting the anti-leukemic activity of blinatumomab in pediatric R/R. Among 113 children who received blinatumomab treatment, via an expanded access program (EAP), 38 of 72 patients in the R/R group achieved a hematological response within two cycles of blinatumomab. Fifty percent of these patients underwent a transplant without bridging myelosuppressive therapy. In patients with evaluable MRD (*n* = 36), 83% (*n* = 30) achieved an MRD response. Taken together, these results demonstrate that the real-world effectiveness of blinatumomab in this cohort of patients was similar to that demonstrated in clinical studies [107]. In addition, the Italian real-life multicenter retrospective study on 39 R/R BCP-ALL pediatric patients (0–21 years old) treated in seven National Cooperative Pediatric Oncology Group (AIEOP) centers represents a further demonstration of blinatumomab’s efficacy (CR rate 46% in 13 patients) and good tolerability (34.8% grade ≥3 AE rates, no CRS and no associated toxic deaths) [108].

### 4.2. Antibody–Drug Conjugates (ADCs)

ADCs are obtained by combining a mAb with a cytotoxic drug using various linkers that determine how and when the drug is detached from the mAb [109]. Following the binding of the surface antigen, ADCs are internalized into the endosomes and then released in the lysosomes. Finally, when ADCs are delivered to the nucleus, they induce cell death [110,111]. It has also been shown that non-internalized ADCs directed against the tumor microenvironment (TME) components can efficiently liberate their drug in the extracellular space and mediate a potent therapeutic activity (bystander killing effect) [112] (Figure 2).

CD22 is expressed in around 90% of B-ALL cases and can be considered an ideal B cell target for immunoconjugate treatment. Inotuzumab ozogamicin (InO) is a humanized CD22 monoclonal antibody conjugated to calicheamicin [113], approved for R/R ALL adult patients’ treatment. Its approval was supported by the results of the global open-label phase 3 randomized INO-VATE study (NCT01564784) that investigated the efficacy and safety of InO vs. chemotherapy in R/R ALL patients with low, moderate, or high disease burden. Patients treated with InO had improved CR rates and improved median OS than those treated with chemotherapy (CR rates: 81% vs. 29%; OS: 7.7 vs. 6.7 months) [113].

Moreover, DeAngelo et al. carried out a post hoc analysis of INO-VATE to evaluate the long-term efficacy and safety profile of InO also in adult patients R/R-ALL with high baseline disease burden. They reported comparable clinically meaningful benefits to those observed in subgroups with low and moderate disease burden [114]. However, patients receiving a starting dose of 1.8 mg/m^2^/cycle (in three divided doses) of InO had an increased risk of sinusoidal obstruction syndrome (SOS), especially following HSCT, compared with those who received standard chemotherapy (13% vs. <1%) [113]. Therefore, other groups are evaluating how to prevent SOS. One of the strategies (NCT03677596) is based on the use of a lower dose of InO (1.2 mg/m^2^/cycle). Initial results showed that half of the patients (11/22) achieved remission, and more than 70% of them achieved MRD negativity [115]. Additional studies recommend ursodiol prophylaxis, limited use of InO at no more than two cycles before HCT, and no dual alkylating agents such as thiotepa and melphalan and hepatotoxic agents [116].

Despite the good results of InO treatment in adult ALL, there is still little information on its safety and efficacy in childhood ALL. However, available data from European and American compassionate use programs demonstrated a favorable benefit–risk profile of InO in children with R/R BCP-ALL. To provide more comprehensive data on InO effectiveness and tolerability, phase 2 prospective studies on pediatric patients with R/R ALL are currently underway in the United States (NCT02981628/AALL1621) and Europe (EudraCT 2016-000227-71). Results of the Children’s Oncology Group trial AALL1621 on InO treatment in children and adolescents with R/R B-ALL were presented at Asco 2022. In cycle 1, patients received a starting dose of InO of 0.8 mg/m^2^ intravenously on day 1 and 0.5 mg/m^2^ on days 8 and 15 of a 28-day cycle with response evaluation on day 28. Dose-limiting toxicities and SOS were continuously checked. Nineteen out of forty-eight patients had CR and nine had CRi after cycle 1. Twenty-one patients received HSCT after InO, of whom six developed grade 3 SOS. InO was effective with high response rates and MRD < 0.01% in two-thirds of responders even if SOS after HSCT and prolonged cytopenias were notable [117].

Furthermore, the phase 1 trial (ITCC-059), registered as EudraCT 2016-000227-71, investigated the recommended phase 2 dose of InO in children with multiple R/R ALL. Twenty-five patients aged 1 to 18 years were enrolled in the study and, among these, twenty-three were included in the dose escalation analysis (three doses of InO per course). Severe adverse events were observed in 23 patients and hepatic SOS was evidenced in 2 patients following chemotherapy. OR after one course was achieved in 20 of 25 patients, of whom 84% reached CR-MRD negative. The OR at 12 months was 40%. The recommended phase 2 dose (RP2D) of InO was established at 1.8 mg/m^2^ per course for adults [118]. Such results demonstrate the efficacy of these antibody constructs and support new design approaches based on the synergistic potential of either or both agents with low-intensity chemotherapy to further improve outcomes, especially in older patients. The first data obtained indicate that the use of InO in combination with low-intensity chemotherapy (mini-hyper-CVD) with or without blinatumomab confers better outcomes than standard intensive chemotherapy (hyper-CVAD)as first-line therapy in adult patients with ALL [119,120,121,122,123].

### 4.3. Chimeric Antigen Receptor (CAR)-Engineered Immune Cell Therapy in ALL

In this section, we review one of the most promising immunotherapy approaches for ALL, consisting of the genetic modification of immune cells such as T cells and also NK cells with chimeric antigen receptors (CARs). It is known that the immune system plays a crucial role in tumor growth control. The tumor, however, may escape detection by the immune system, and its growth and progression are controlled by TME. Hypoxia, also as a consequence of ischemia, and nutrient deprivation are only some of the ways used by TME to destabilize immune cells. Hypoxia can shape the type and function of immune cell infiltration in the TME by polarizing tumor-associated macrophages (TMAs) toward anti-inflammatory M2 macrophages and cytokines [124,125,126,127]. Furthermore, immune cell dysfunction is mediated by a series of factors including the changes in signal transduction molecules, loss of TSA, stimulation of CTLA4 on T cells, and secretion of some soluble molecules by tumor or non-tumor cells in the TME, other than by the presence of some immunosuppressive cells in TME [128,129]. In this context, the engineering of CART cells has become the new frontier of immunotherapy in the treatment of hematological malignancies, even if it has important adverse events limiting its success. CRS and neurotoxicity together to on-target off-tumor effects and GVHD are only some of the restrictions linked to a broader application of CART cell therapy in hematological disease treatment. To overcome these limitations, other immune effector cells that may be modified with CARs and used in immunotherapy are being studied. The scientific focus has recently shifted to NK cells, whose particular molecular peculiarities make them suitable for an “off-the-shelf” allogeneic therapy. First, it is possible to produce big NK cell quantities from several sources, and this, together with a minimal risk of toxicity or GVHD permitted by the HLA-I dominant-negative regulatory function on NK killing activity and a minor cost of production, makes CARNK cell therapy the most promising immunotherapy for leukemia.

#### 4.3.1. CART Cells

CART cell therapy is based on the genetic engineering of a patient’s T cells to induce the expression of a chimeric receptor able to recognize a marker expressed on tumor cells leading to cancer cell elimination. T cells are sampled from the patient’s peripheral blood and transduced with viral vectors encoding the desired genes. Genetically engineered cells are then expanded in vitro before re-infusion into the patient’s blood (Figure 3).

Currently, CART cells can be categorized into four generations based on the organization of their intracellular signaling domain, with fifth-generation CARs similar to the second-generation, but with an intracellular domain of a cytokine receptor [130,131,132,133,134,135].

CD19 is an ideal target antigen for CART cell therapy, and encouraging results have been reported in the treatment of several types of B-cell malignancies [136]. In 2013, for the first time, CD19-directed CART (CART 19) cells were successfully used in two children with chemotherapy-resistant ALL, and despite the presence of severe CRS and B-aplasia, CR was achieved in both patients [137]. Since then, several investigations have been carried out to better understand the CR rate and the durability of the CART cell therapy effect, and early reports demonstrated the potential benefits of CAR T cells in R/R B-ALL [138,139,140,141,142,143].

In 2018, FDA approved the anti-CD19 CART cell therapy tisagenlecleucel (CTL019) for R/R B-ALL based on the results of the ELIANA multicenter study (ClinicalTrials.gov number, new cases and 1560 deaths (incluNCT02435849), which showed high response rates in patients up to 25 years of age. Although transient high-grade toxic effects occurred, an overall remission rate of 81% among 75 patients at 3 months of follow-up after a single infusion of tisagenlecleucel was reported [138].

In the same year, Park et al. used a CD19 CART construct with a CD28 costimulatory domain (19–28z) in a phase 1 trial on 53 adults with relapsed B-cell ALL. They hypothesized that the safety and long-term efficacy of 19–28z CART cells may be associated with the clinical characteristics of the patients, disease characteristics, the treatment regimen, and the kinetics of T-cell expansion. They reported that 14 out of 53 patients developed severe CRS and 1 patient died. CR was achieved in 83% of the patients and MRD response was observed in 67%. The median OS was 12.9 months. Moreover, 19–28z CART cell therapy (median follow-up of 29 months) showed a favorable long-term remission rate in patients with a low disease burden, who had significantly longer event-free survival and OS with a markedly lower incidence of toxic effects than did those with a high disease burden [140].

KTE-X19, another anti-CD19 CART cell therapy, already approved for non-Hodgkin lymphoma, has also been studied for the treatment of B-ALL. Data from the ZUMA-3 trial, which was conducted on adult patients with R/R B-ALL, showed a high response rate and tolerable safety of KTE-X19.Fifty-four patients were enrolled in phase 2 of the trial and forty-five of them received a single infusion of the CD19-directed product (2 × 10^6^, 1 × 10^6^, or 0.5 × 10^6^ cells per kg) after lymphodepleting chemotherapy. Severe CRS and neurotoxicity, which occurred in 31% and 38% of the patients, respectively, were successfully managed. CR was achieved in 52% of patients within 3 months [144]. Furthermore, at the 2021 American Society of Clinical Oncology (ASCO) Annual Meeting, Shah et al. presented the results of the phase 2 portion of this trial, reporting that the CR/CRi rate was 71%. After a median follow-up of 16.4 months, KTE-X19 showed compelling clinical benefit in heavily pretreated adults with R/R B-ALL, with the median OS not yet reached for responding patients and a manageable safety profile [145]. Finally, in October 2021, KTE-X19 was approved as the first CART cell therapy for adults with R/R B-ALL.

Interestingly, CART cells can migrate to CNS or testes and thus can be considered a good therapeutic choice also for the treatment of CNS-relapsed leukemia [146,147,148]. In 2015, Rheingold et al. demonstrated that CTL019 was detectable in cerebrospinal fluid in 46 out of 47 treated patients with B-ALL, indicating the ability of this therapy to cross the blood–brain barrier [149].To date, only a few results are available on the efficacy of CART cell therapy in patients with R/R B ALL and active CNS disease [141,150,151]. However, data from the CHP959 trial (NC01626495), carried out on 65 patients with CNS involvement, showed no significant differences in relapse-free survival or neurological toxicities between patients with active CNS disease and those without it, before CD19 CART cell infusion [152]. Other studies confirmed the efficacy of this treatment in patients with multiply relapsed or refractory extramedullary leukemia [153,154].

The value of CART cells is undeniable in the treatment of B-ALL, but it is necessary to understand how to minimize toxicities such as CRS, immune effector cell-associated neurotoxicity syndrome (ICANS), and B-cell aplasia related to it, particularly in adult patients. CRS is a systemic inflammatory response that is often associated with CART cell therapy within 1–4 days after the infusion and can progress to multiple organ dysfunction. Learning how to recognize early CRS is a fundamental step to treating it promptly and preserving life-threatening consequences. Of note, higher-grade CRS has been associated with higher disease burden and may be effectively treated with the anti-interleukin-6 receptor antibody tocilizumab which, however, could limit the efficacy of the immunotherapy [155,156]. In addition, a relationship between CART dose and CRS occurrence has been highlighted [157]. Therefore, the adoption of a fractionated dosing scheme might be a good strategy to retain high response rates with acceptable tolerability. Meaningful advancement was shown by Frey’s group which demonstrated that fractionation of CTL019 dosing treatment can help manage CRS toxicity and maintain efficacy in adults with R/R ALL [158]. ICANS is also associated with CART cell therapies and it seems due to both activated CART and endogenous T lymphocytes and the cytokines secreted by them [159,160,161]. It can occur in association with or following CRS, and its management continues to evolve and constitutes an area of ongoing research. In addition, B-cell aplasia represents another CART cell-related toxicity linked to CD19 CART cell therapy for B-ALL [162]. Hypogammaglobulinemia and agammaglobulinemia caused by B cell aplasia expose patients to an increased risk of infection that needs to be promptly managed to avoid lethal consequences [162,163]. To this aim, immunoglobulin replacement may help to prevent serious bacterial infections [138,164,165,166,167]. It has been demonstrated that increasing serum IgG levels may result in protection against infections [162]. Moreover, antimicrobial and antifungal prophylaxis is also recommended in the prevention of infections in patients treated with CD19-redirected CART cell therapy [168,169,170]. In case of viral infections such as herpes simplex virus and varicella zoster virus reactivation, following CD19 redirected CART therapy, antiviral prophylaxis should be considered. Already in the past, researchers were interested in studying the possible impacts of immunotherapy in leukemic patients with viral infections such as HIV and hepatitis B and C. Until the past few years, in the presence of viral infections, patients were not considered for immunotherapy treatment that could worsen the infection.

However, during the COVID-19 pandemic, studies on the interaction between the immune system and acute respiratory syndrome coronavirus 2 (SARS-CoV-2) indicated that the coronavirus can promote PD-L1 expression, towards which several immunotherapy drugs are directed. Therefore, in the case of coronavirus infection, in the early phases, an immunotherapy regimen could have positive effects on counteracting the virus by stimulating the patient’s immune system against it. However, in severe SARS-CoV-2, the use of immunotherapies could represent a risk for the inflammatory storm associated with a hyperactive inflammatory response. Recently, a prolonged severe SARS-CoV-2 infection in a BCMA-redirected CART cell therapy recipient was described. Despite convalescent plasma therapy and antiviral prophylaxis with the agent Remdesivir, the patient experienced a massive lung infection and died from infection-related complications [171].

#### 4.3.2. CARNK Cell Therapy

As depicted in Figure 4, NK cell activity is controlled by multiple inhibitory or stimulating receptor-ligand interactions depending on a health condition or disease [172].

NK cells are heterogenous and distinct cell subsets mediating specialized functions. The tissue of origin of NK cells is the bone marrow in which IL15 and, to a lesser extent, IL2 play a pivotal role in NK cell development and differentiation [173,174]. Among NK cells, two cell subsets emerge in terms of cell function. The first subset is composed of the classic cytotoxic cells and the second subset is composed of NK cells with regulatory functions. Both NK cell subsets are defined according to the intensity of cell surface expression of CD56 and CD16. The first includes CD56^low^CD16^high^ cells and the second is characterized by CD56^high^CD16^low/neg^ cells. The CD56^low^CD16^high^ are professional killer cells. NK cell cytotoxicity is regulated by a balance between activator and inhibitory molecules termed natural cytotoxic receptors and killer inhibitory receptors (KIRs), respectively (missing-cell hypothesis) (Figure 4) [175].The role of NK cells in the host defense against solid tumors is unclear. However, there is evidence that NK cells may play a minimal direct role in counteracting epithelial cancers, but they can cooperate with T cells in controlling tumor progression. For example, NK cells are barely found in the TME, and even if they are found, this may not be associated with improved survival [176,177,178]. However, pre-clinical and clinical studies have shown that NK cells play a pivotal role in the immune response against leukemia in allogeneic, HLA-matched, and unmatched settings. Unlike T cells, they do not mediate GVHD. As a consequence, NK cells are interesting effector cells for cell-based immunotherapy for leukemias [179,180].

NK cells can be obtained from several sources, including donor or autologous peripheral blood mononuclear cells (PBMCs), umbilical cord (UCB), cell lines (NK-92), pluripotent stem cells (PSCs) from human embryonic stem cells (hESCs), and induced pluripotent stem cells (iPSCs). All these sources of NK cells can, subsequently, be engineered with a CAR, expanded, and infused into the patient (Figure 5).

Analyzing and comparing the efficacy of methods for NK cell engineering, it has been shown that NK cells can be quickly isolated from peripheral blood (PB). However, they are difficult to engineer due to low transduction efficiency combined with poor expansion. Instead, NK-92 cells demonstrate a strong anti-tumor activity [181] that makes them a good option for engineering, even if they need to be irradiated before use to prevent lethal effects. To overcome the limitations of long-term storage decreasing the cytotoxic capabilities, a good choice is UCB-derived NK cells. Indeed, these cells may undergo cryopreservation with minimal alterations, and despite their relatively immature nature, exhibit high proliferative capabilities and work effectively for in vivo studies compared to PB-derived NK cells [181]. Moreover, manufacturing iPSC-NK cells may be considered a good alternative. IPSC-NK cells are quick to obtain, safe, and show high cytotoxic activity against tumor cells. To further improve the efficacy of CARNK cells, several gene-editing strategies to enhance their potential, their persistence, and homing are being studied [182].

NK cell-mediated immunotherapy is based on increased NK cell activation via blocking inhibitory interactions, expanding NK cell populations, and improving overall function. Although it is still in the experimental phase, its potential is amply suggested by longer survival and reduced relapse together with fewer adverse effects than CART cell therapies [183].

Initial studies on CARNK cells have been initiated by CART cell constructs since NK cells and T cells share some costimulatory domains such as 4-1BB. However, other co-stimulatory domains, more specific for NK cell signaling, are being investigated. In particular, NKG2D and CD244 (2B4) are the two costimulatory molecules through which NK cells raise their cytotoxic capability and cytokine production [184]. Early results from ongoing clinical trials are encouraging and demonstrate that NK cells provide a safer and more advantageous CAR-engineering platform than T cells [185]. This permits us to hypothesize that a large number of patients can be treated on demand with this new immunotherapy. However, to date, only a few clinical studies of CARNK cell immunotherapy for ALL patients are going on. This may be partially due to the modest outcomes obtained from the use of first-generation CARNK cells [186]. Among the most promising clinical trials, there is NCT05020678, a single-arm, open-label, multicenter, phase 1 study that is ongoing to evaluate the safety and tolerability of an experimental intravenous allogeneic CARNK cell targeting CD19 (NKX019) in adult patients (*n* = 60) with relapsed/refractory NHL, CLL or B-ALL. NCT05563545, a single-arm clinical study, is instead recruiting cases with recurrent or refractory CD19 positive ALL to evaluate the safety, dose, tolerance, and pharmacokinetic characteristics of CARNK-CD19 (SNC103) and also define the effectiveness, the immunogenicity of the product, and the correlation between the changes in cytokines after infusion and CRS and ICANS. Furthermore, in NCT04796688, patients with CD19+ R/R ALL are being recruited to be treated with fludarabine + cyclophosphamide + CARNK-CD19 cells and evaluate the safety and efficacy of universal CAR-modified AT19 cells. Instead, in NCT04796675, NK cells derived from healthy donor cord blood (CB) have been engineered with an anti-CD19 CAR to test their safety and efficacy in patients with CD19+ B cell malignancies. Additionally, in the active but still not recruiting NCT03056339 clinical study, CB-derived NK cells are being used. The purpose of the study is to learn if iC9/CAR.19/IL15-transduced CB-NK cell infusion, after fludarabine and cyclophosphamide associated with mesna, improves the disease in patients with R/R B-cell leukemia. Another goal of the study is to find the highest tolerable dose of CARNK cells to give to patients and evaluate the safety of this treatment. Among the several NK lines, the NK-92 cell line has been successfully modified to express CARs recognizing antigens expressed on tumor cells and may be considered an ideal source for cell-based immunotherapy. A previous phase 1 clinical study demonstrated that NK-92 cells can be irradiated at very high doses with minimal toxicity in patients with refractory hematologic tumors, who had relapsed after autologous hematopoietic cell transplantation [187]. The clinical trial NCT02892695, started in 2016, is one of the first clinical trials with engineered NK-92 cells for CAR therapy. Ten patients with leukemia (including ALL) or lymphoma have been enrolled to evaluate the safety and optimal dose of CARNK PCAR-119 used as bridge immunotherapy before receiving stem cell transplantation. In the ongoing NCT02727803 phase 2 study, CAR-engineered NK-92 cells are used in patients (estimated enrollment of 100 patients with myelodysplastic syndrome, leukemia, lymphoma, or multiple myeloma) that have received cord blood transplantation.

The primary objective of the study is to define the progression-free survival (PFS) time and then evaluate OS time for treatment-related mortality (TRM) and adverse events (GVHD/infection). In addition to CD19, another promising target for CARNK cell therapy is CD7. In NCT02742727, the NK-92 cell line has been engineered to express an anti-CD7 attached to TCR zeta, CD28, and 4-1BB signaling domains and to be infused in patients with CD7-positiveR/R leukemia and lymphoma to evaluate its safety and effectiveness. The NCT02890758 clinical trial is underway to investigate the number of NK cells from non-HLA matched donors (this kind of infusion is still experimental and not approved by the FDA) that can be safely infused into patients with hematologic tumors. After receiving the NK cells, patients may also be given ALT803, a drug that keeps NK cells alive, promotes their expansion, and supports their cancer-fighting characteristics. Furthermore, to enhance the therapeutic utility of NK-92 cells for the treatment of B-ALL, Oelsner et al. engineered NK-92 cells with an FMS-like tyrosine kinase 3 (FLT3)-specific CAR containing a composite CD28-CD3ζ signaling domain. Their results suggest that FLT3-specific CAR NK cells exhibit high and selective cytotoxic activity against established and primary B-ALL cells in vitro, and in a NOD/SCID IL2Rγ-null mouse xenograft model of B-ALL, a remarkable inhibition of disease progression was observed, thus demonstrating high antileukemic activity in vivo [188].

## 5. New Treatments under Investigation

Future research directions aim to minimize chemotherapy and HSCT and to improve the life expectancy of patients with ALL, especially older patients and those with ALL resistant to available treatments. The question is what real advances in ALL therapy can we expect in the next future? The answer will come from the use of combination therapies capable of reducing drug resistance and improving drug efficacy to obtain a better and longer outcome. Therefore, several studies are currently ongoing to evaluate different drug combination uses in relapse and frontline treatment settings.

In Ph+ ALL, the combination of blinatumomab with TKI, particularly the third-generation ponatinib, is showing promising efficacy, with a deep and durable response and less need for both chemotherapy and HCST in the first remission [189].

The results of a phase 2 monocentric study presented at ASH 2021 by Short et al. evidenced that the combination of these two agents had synergistic effects on apoptosis. While ponatinib inhibits BCR-ABL kinases, blinatumomab promotes an antitumor response against CD19-expressing B cells [190].

This chemotherapy-free combination of ponatinib and blinatumomab was safe and effective in both newly diagnosed(ND) and R/R Ph+ ALL patients. Particularly favorable outcomes (estimated 2-year event-free survival (EFS) and OS 95%) were reported for the ND cohort that was not transplanted in first remission, suggesting that this regimen may serve as an effective transplant-sparing therapy in these patients [190]. Good results were also obtained by combining ponatinib with lower-intensity chemotherapy (hyper-CVAD) as an initial treatment for adult patients (age ≥ 18–75 years) with ND-Ph+ ALL (NCT01424982). Stable, long-term remission has been shown in 70% of patients [191]. Other encouraging data were presented by a Spanish group (PETHEMA) that carried out a phase 2 PONALFIL trial, in which ponatinib (30 mg/d) was combined with an induction/consolidation chemotherapy followed by HSCT to treat adults with ND-Ph+ ALL [192]. In comparison to a more conventional therapeutic approach, this combination therapy showed good clinical activity and a favorable toxicity profile. CR was achieved in 100% of patients (30/30), 14 (47%) of whom obtained CMR and 5 (17%) MMR.

Induction with TKIs is showing promising results also in the phase 3 PhALLCON study(NCT03589326), where a comparison of first-line ponatinib (PON) vs. imatinib (IM) with reduced-intensity chemotherapy (CT) has been carried out in patients with newly diagnosed Ph+ ALL. The first report of PhALLCON demonstrates that PON resulted in more durable and deeper responses, with a trend toward improved EFS and comparable safety vs. IM.

With regard to CART cell therapy, given the success of tisagenlecleucel (Kymriah), approved for use in children and young adults up to the age of 25, and more recently of brexucabtagene autoleucel (Tecartus), approved for all adult patients with R/R B-ALL, future works are focusing on designing new CAR structures with improved anti-tumor efficacy and a better safety profile.

Among multiple strategies, there is one in the phase 1 ALLCAR19 study (NCT02935257), which evaluated the effectiveness of a novel CD19 CAR that uses non-mobilized autologous leukapheresis (CAT-41BBzCAR, also known as AUTO1, another name: obecabtagene autoleucel, obe-cel) in adult patients with R/R B-ALL. Updated data showed a tolerable safety profile of AUTO1 in adult patients with R/R B-ALL despite the high disease burden [193,194].Furthermore, another phase 1b/2 trial (FELIX trial—NCT04404660) aims to find the best balance between the safety and efficacy of this CAR construct. Obe-cel has a lower affinity for CD19 than similar CART cell products and this helps to avoid CART cell over-activation and exhaustion so the T cells can stay active for a longer period [195].The first results by Roddie et al. demonstrated that this therapy has a good safety profile and high remission rates and it might offer a durable treatment option for these patients [194].

To overcome frequent relapses (10–20% of patients) following CD19 CART therapy, a CD22 CART cell therapy has been developed. Pan et al. demonstrated that CD22 CART cell therapy was highly effective in inducing remission in R/R B-ALL patients who failed from previous CD19 CART cell therapy and can be also considered a good bridge for subsequent transplantation to achieve durable remission [196]. Another useful strategy seems to be the development of dual-target CARs by simultaneously targeting CD19 and a second antigen (CD22 or CD20). As shown by Dai et al., this approach is feasible, safe, and able to induce remission in adult patients with R/R B-ALL [197].

Progress in targeted immunotherapies for B-ALL also generated big expectations for T-ALL therapy. Glucocorticoids (GCs) represent central components of T-ALL therapy, and the early response to GC-based therapy is an important predictor of long-term outcomes [198]. Nevertheless, relapse continues to represent a challenge in the clinical management of T-ALL. At the moment, nelarabine, a purine deoxyguanosine analog that inhibits DNA synthesis, remains the only drug for treating the relapse of T-ALL (response rates of over 50% in children and 36% in adults) [199,200,201]. Therefore, there is a need to develop efficient methods of augmenting the response to GC and overcoming resistance to steroid treatment. Most of the ongoing preclinical studies involve novel drugs able to enhance the results of glucocorticoid therapy and that could potentially be included in the induction phase in newly diagnosed ALL patients to prevent relapse and provide better outcomes.

Therapies targetingNOTCH1, such as the proteasome inhibitor (bortezomib) [202,203], JAK inhibitors (ruxolitinib) [204], BCL inhibitors (venetoclax) [205], and anti-CD38 therapy (daratumumab) [206], are showing promising results for better prognosis of patients with T-ALL.

## 6. Conclusions

Immunotherapy revolutionized the treatment of ALL permitting the achievement of remarkably effective and durable clinical responses. However, there is still a significant subset of patients who do not benefit from it. Therefore, research is now focusing on understanding the mechanisms of immune evasion employed by leukemia cells for developing novel therapeutic strategies. This will help investigators understand which patients will benefit the most from immunotherapeutic approaches.

## Figures and Tables

**Figure 1 cancers-15-03346-f001:**
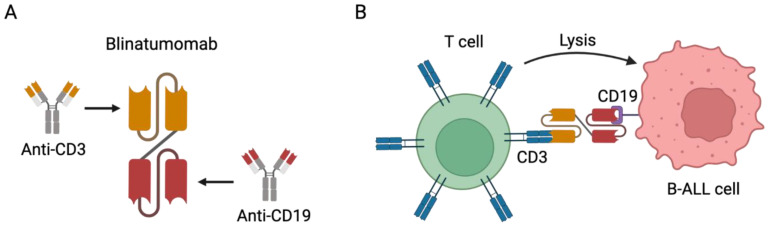
Blinatumomab immunotherapy for the treatment of CD19-positive B cell precursor ALL. (**A**) Representative scheme depicting the structure of blinatumomab. It comprises the fusion of the scFv of CD3 to the scFv of CD19 with a linker. (**B**) Representative scheme depicting the mechanism of action of blinatumomab. It can specifically bind both the CD3 expressed on T cells and the CD19 expressed on B-ALL cells, cross-linking the T cells and B-ALL cells and mediating cell lysis. Note: Figure created with BioRender.com.

**Figure 2 cancers-15-03346-f002:**
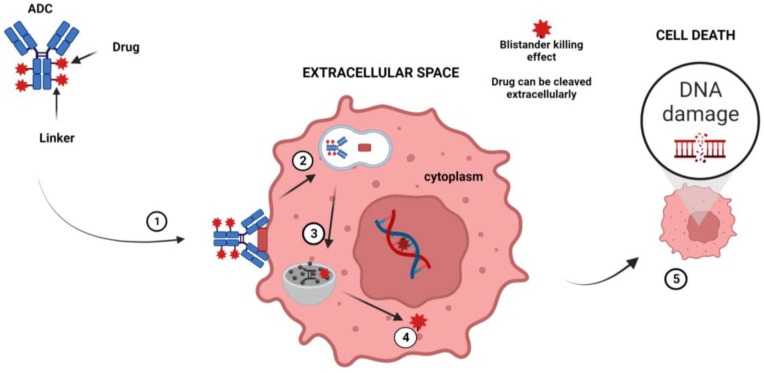
Mechanism of action of ADC. The principal steps of the mechanism of action are as follows: (1) The ADC binds to a tumor-cell-specific antigen, (2) the ADC–antigen becomes a coated pit vesicle and undergoes endocytosis, (3) the ADC is degraded by lysosomal proteases, (4) the drug is released into the cytoplasm, and (5) the drug disrupts the DNA of the targeted cell, leading to cell death. Note: Figure created with BioRender.com.

**Figure 3 cancers-15-03346-f003:**
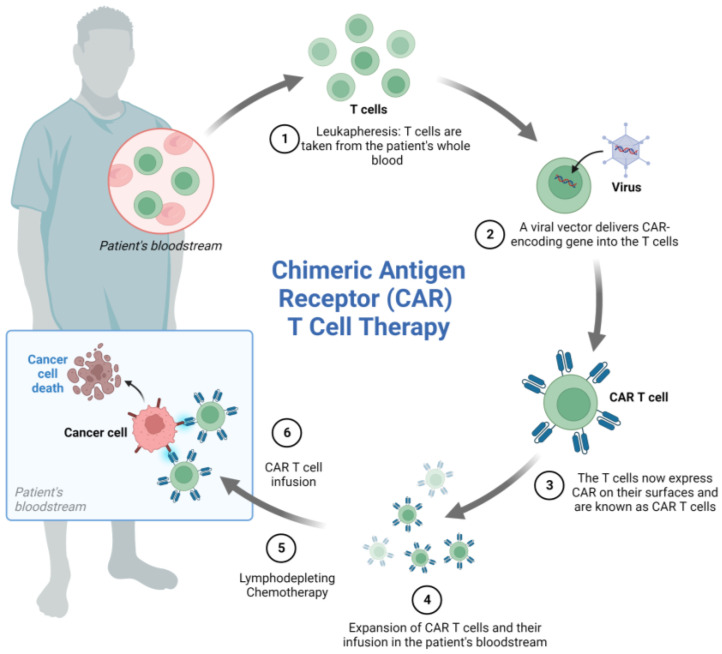
CART cell generation and mechanism of action. The process of CART cell therapy includes the following steps: (1) The patient’s T cells are collected by leukapheresis, (2) a viral vector delivers a gene encoding the CAR into the T cells, (3) the T cells start expressing the CAR on their surface, (4) the CART cells are expanded, (5) the patient undergoes lymphodepleting chemotherapy prior to receiving a CAR-T cell infusion, (6) CART cells infused back into the patient’s blood can attack and destroy cancer cells. Note: Figure created with BioRender.com.

**Figure 4 cancers-15-03346-f004:**
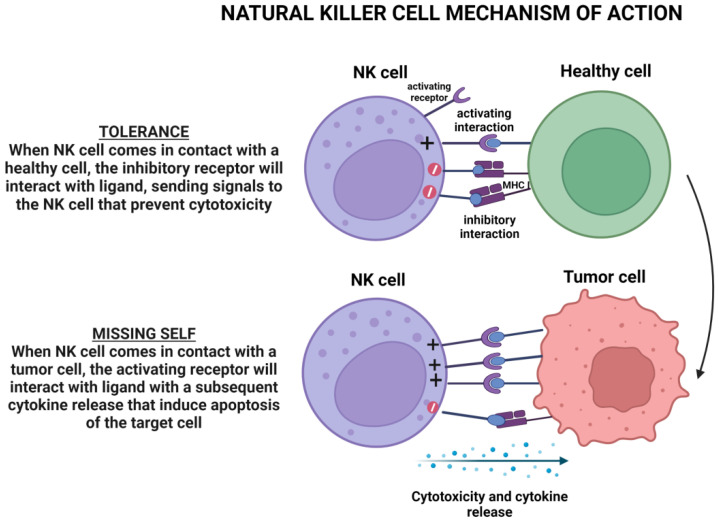
NK cell activity. The engagement of NK cell inhibitory or activating receptor signaling decides the outcome of the immune synapse. Note: Figure created with BioRender.com.

**Figure 5 cancers-15-03346-f005:**
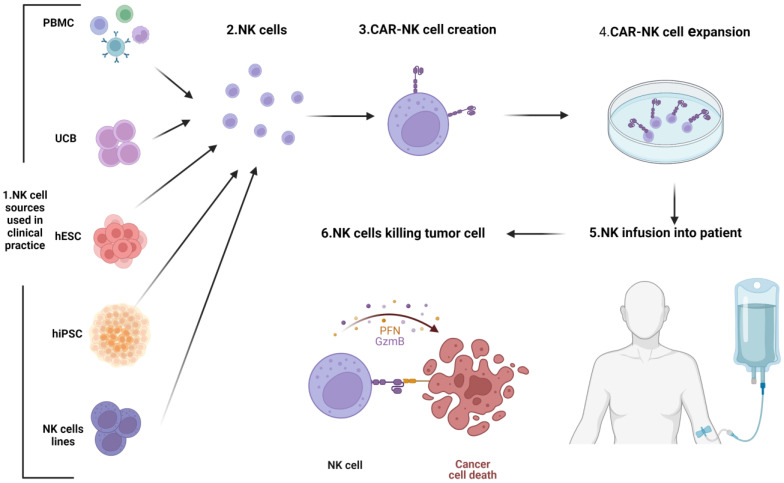
Novel CARNK cell therapy. Representative scheme depicting the process of CARNK cell generation for clinical use. Various cellular sources are utilized for the isolation or differentiation of NK cells. The NK cells are then engineered to express a CAR on their surface and expanded in culture. Following ex vivo expansion, the CARNK cells are infused into the patient and are re-directed to target and destroy cancer cells. Note: Figure created with BioRender.com.

**Table 1 cancers-15-03346-t001:** WHO classification of acute lymphoblastic leukemia.

**B-lymphoblasticleukemia/lymphoma**
B-lymphoblasticleukemia/lymphoma, NOS
B-lymphoblastic leukemia/lymphoma with recurrent genetic abnormalities
B-lymphoblastic leukemia/lymphoma with t(9;22)(q34.1;q11.2);BCR-ABL1
B-lymphoblastic leukemia/lymphoma with t(v;11q23.3);KMT2A rearranged
B-lymphoblastic leukemia/lymphoma with t(12;21)(p13.2;q22.1);ETV6-RUNX1
B-lymphoblastic leukemia/lymphoma with hyperdiploidy
B-lymphoblastic leukemia/lymphoma with hypodiploidy
B-lymphoblastic leukemia/lymphoma with t(5;14)(q31.1;q32.3)IL3-IGH
B-lymphoblastic leukemia/lymphoma with t(1;19)(q23;p13.3);TCF3-PBX1
Provisional entity:B-lymphoblastic leukemia/lymphoma with translocations involving tyrosine kinases or cytokine receptors (“BCR-ABL1–like”)
Provisional entity:B-lymphoblastic leukemia/lymphoma with intrachromosomal amplification of chromosome 21 (iAMP21)
**T-lymphoblastic leukemia/lymphoma** (can only be differentiated from B-ALL/LBL based on IHC and/or flow cytometry).
Provisional entity: Early T-cell precursor lymphoblastic leukemia
Provisional entity: NK cell lymphoblastic leukemia/lymphoma

**Table 2 cancers-15-03346-t002:** Blinatumomab in pediatric B-lineage ALL. Clinical trials of blinatumomab are reported by year, country, and population of pediatric patients with B-lineage ALL.

Publication Year (Ref)	Participating Countries	Patient Selection
2014 [82]	Germany	9 R/R-BCP-ALL patients post-HSCT
2016 [83]	26 European and US Centers	70 R/R-ALL patients (out of 93) who received the recommended dose of blinatumomab 5/15 µg/m^2^/day
2017 [84]	Czech Republic, US, Canada, France, Germany, Italy	18 BCP-ALL patients (4 with CD19-negative relapse)
2017 [85]	Germany	1 relapsed ALL patient without MLL rearrangement (case report)
2018 [86]	US-Birmingham, Alabama	1 BCP-ALL patient with Down syndrome (case report)
2018 [87]	26 European and US Centers	70 R/R-ALL patients—follow-up study
2018 [88]	Germany	1 ALL patient without MLL rearrangement(case report)
2019 [89]	Israeli	11 BCP-ALL patients with overwhelming toxicity
2019 [90]	US	15 R/R-ALL patients with residue MRD
2019 [91]	European experience from International BFM Study group	9 B-ALL patients with t(17;19)(q22;p13)/TCF3-HLF
2019 [92]	US, Austria, Canada, France, Germany, Italy, Netherlands	59 R/R BCP-ALL patients (MT103-205 single-arm multicenter phase 2 study)
2020 [93]	US, Austria, Canada, France, Germany, Italy, Netherlands	70 R/R Ph-BCP-ALL patients (MT103-205 single-arm multicenter phase 2 study)—blinatumomab vs. standard therapy
2020 [94]	US, Austria, France, Germany, Italy, Switzerland, UK	110 R/R ALL patients—RIALTO expanded access study
2020 [95]	UK, Ireland	11 infants with persistent MRD
2020 [96]	Greece	9 R/R-ALL patients
2020 [97]	Japan	9 R/R-ALL patients
2020 [98]	Russia	90 R/R-BCP-ALL patients
2021 [99]	Spain	27 R/R B-ALL patients (children/AYA)
2021 [100]	France	1 infant with KMT2A rearranged ALL (case report)
2021 [101]	Germany	38 R/R BCP-ALL patients
2021 [102]	Australia	24 R/R BCP-ALL patients
2021 [103]	US, Canada, Australia, New Zealand	208 first relapsed B-ALL patients (aged from 1 to 30 years)
2021 [104]	Europe, Australia, Israeli	108 first relapsed B-ALL patients(aged 28 days to 18 years)
2022 [105]	US, Austria, France, Germany, Italy, Switzerland, UK	110 patients RIALTO expanded access study-FINAL ANALYSIS
2022 [106]	France, Italy, Russia, Spain, United Kingdom	72 with R/R Ph−BCP-ALL and 41 with MRD+, either Ph−or Ph+:retrospective observational study
2022 [107]	Italy	39 R/R ALL patients: real-life multicenter retrospective study in 7 AIEOP Centers

HSCT, hematopoietic stem cell transplantation; R/R ALL, relapsed or refractory acute lymphoblastic leukemia; BCP-ALLB-Cell Precursor, acute lymphoblastic leukemia; MRD, minimal residual disease; MLL, mixed-lineage leukemia; Ph+, Philadelphia chromosome-positive; Ph−, Philadelphia chromosome-negative; KMT2A, histone–lysine N-methyltransferase 2A.

## Data Availability

Not applicable.

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
