# Peer review of "Acute Lymphoblastic Leukemia Immunotherapy Treatment: Now, Next, and Beyond"

_cancers, 2023, doi:10.3390/cancers15133346_

Round 1

Reviewer 1 Report

The review article provides a concise overview of ALL, highlighting its prevalence in children and adults, the involvement of different cell lineages, and the challenges associated with treatment outcomes. The figures are very well created and the writing is easy to understand to a broader audience. There are certain spelling/spacing errors in line, 23,42,430, and 470 that needs to be corrected.  

Author Response

We thank the Reviewer for having appreciated our review and for the positive comments. According to her/his suggestions, we provided to correct spelling/spacing errors.

Reviewer 2 Report

In this review, Aureli and colleagues do a nice state of the art analysis of the use of immunotherapy in ALL. The text is understandable, well-presented and appropriately referenced. I only have 3 minor comments shown below.

- WHO 2022 (Alaggio Leukemia 2022) should be referenced instead that of 2016. In addition, ICC 2022 (Arber Blood 2022) should also be mentioned.

- In lines 305-306 (CARNK) authors could be not so categorical and could introduce i.e. “may” or “the most promising”

- In line 422 “leaukemias” should be corrected.

Author Response

We thank the Reviewer for the positive comments; we embraced all his/her suggestions.

Reviewer 3 Report

The authors have systematically and clearly summarized immunotherapy treatment for acute lymphocytic leukemia.

I believe the following additional information, if possible, would be even more useful to the reader.

1.       Blinatumomab added to chemotherapy in infant acute lymphoblastic leukemia. (NEJM 2023; 388:1572-1581)

2.       How many courses of blinatumomab therapy is appropriate? Do you think blinatumomab should be administered after allogeneic transplantation? Please state as much as you can in the literature.

3.       Please add what is known about the differences between the blinatumomab responder group and the blinatumomab non-responder group.

4.       Should allogeneic transplantation be performed after CAR-T treatment in young patients? Please discuss this. (Should CAR-T therapy be the last treatment?)

Author Response

We thank the Reviewer for his/her positive comments that gave us the possibility to improve this section of our manuscript. In pages 5-6 there is an insight about this subject (highlighted in yellow).

Reviewer 4 Report

In this ms., Aureli and colleagues describes very comprehensively the last achievements of the immunotherapy in the treatment of acute lymphoblastic leukemia (ALL). They focused mainly on bispecific antibodies, drug-antibody conjugates and CAR-T/CAR-NK cell therapies. They revised the last findings of the clinical trials using the main immunotherapies and they gave a perspective, in which direction the ALL field is moving.

The manuscript is well written and describes the main findings of the ALL field. The figures are also well done and are very comprehensive. However, there are several aspects, which should be improved/corrected prior publication as described below.

Major points

-       The authors focused the review mainly in the development of immunotherapy, which has undoubtedly improved the outcome of pediatric and adults patients. However, one of the major achievements in the treatment of Ph+ and Ph-like ALL has been the development of more potent TKI (dasatinib, ponatinib, ruxolitinib) in combination with immunotherapy and/or chemotherapy. The authors mentioned the TKI therapy in some parts of the ms. but, I think, they should also emphasize this development in their review with a section focusing on this aspect (in combination with immunotherapy?)

-       The authors write in the ms. the terms “ALL” and “ leukemia” indistinctly. The authors should be very careful because “leukemia” can be many types as myeloid or lymphoblastic, acute and chronic. I think, the correct term to be used in this ms. is “ALL”. Please, revise the ms. accordingly.

Minor points

-       There are many words in the ms. which are come together. Please, revise carefully the entire ms.. For example:

page 1, line 23: affects_children

page 2, line 42:  that_replace -> here, it would be “infiltrate” bone marrow or “displace“ the normal hematopoiesis more accurately.

Page 10, line  317: CART cell_generation

-       Page 2, line 47: please, cite the new WHO 2022 classification instead of 2016

-       Page 6, line 175-178: the authors describe in this section the success of blinatumumab in combination with chemotherapy or TKI but the last sentence is about induction with TKI (ponatinib vs. imatinib). Please, move this last sentence to a different part of the ms.

-       Figure 2, left: the fonts on the table are too small and unreadable. Please show this as a separate table with less information. Probably the map does not add any crucial information to the table and I would recommend to skip it.

-       Figure 4: the lymphodepletion by chemotherapy with f.e. fludarabine/cyclophosphamide is a very important step of the procedure and is missing. Also, the isolation of T cells by leukapheresis is missing. Please, complete the figure.

-       Page 15, line 469: mesna is not a chemotherapy but it is a supportive agent for the cyclophosphamide

-       The authors write in the conclusion that “…single-cell techniques, including massive DNA-RNA sequencing and mass cytometry, will help investigators understand…”. However, the authors does not mention in the ms.  this techniques and how to employ them for the treatment of diagnosis of the ALL. Please, describe in the ms. or delete this part of the conclusion.

n.a.

Author Response

Major points

1: We thank the Reviewer for his/her positive comments. We embraced the suggestions and we provided to expand the discussion about TKI. Notwithstanding this, we preferred not to write an entire section about this topic to not change the overall orientation of the manuscript. We will surely consider this subject for our future studies.

2: We revised the manuscript in accordance to this suggestion.

Minor points

According to the Reviewer’s comments, we have removed the inappropriate expressions and fixed the inaccuracies. We also provided to revise the figures as suggested.

Round 2

Reviewer 3 Report

My previous points were appropriately corrected by the authors.

There are no further points I need to point out.

Reviewer 4 Report

The authors addressed all my comments

Some minor English is required. There are still some words joined together.